# Increased Levels of Circulating IGFBP4 and ANGPTL8 with a Prospective Role in Diabetic Nephropathy

**DOI:** 10.3390/ijms241814244

**Published:** 2023-09-18

**Authors:** Hana Th. AlMajed, Mohamed Abu-Farha, Eman Alshawaf, Sriraman Devarajan, Zahra Alsairafi, Ashraf Elhelaly, Preethi Cherian, Irina Al-Khairi, Hamad Ali, Rose Mol Jose, Thangavel Alphonse Thanaraj, Ebaa Al-Ozairi, Fahd Al-Mulla, Abdulnabi Al Attar, Jehad Abubaker

**Affiliations:** 1Applied Health Science Department, College of Health Sciences, Kuwait 15462, Kuwait; almajed777@hotmail.com; 2Biochemistry and Molecular Biology, Dasman Diabetes Institute, Kuwait 15462, Kuwait; eman.alshawaf@dasmaninstitute.org (E.A.); preethi.cherian@dasmaninstitute.org (P.C.); irina.alkhairi@dasmaninstitute.org (I.A.-K.); 3National Dasman Diabetes Biobank, Dasman Diabetes Institute, Kuwait 15462, Kuwait; sriraman.devarajan@dasmaninstitute.org (S.D.); rose.jose@dasmaninstitute.org (R.M.J.); 4Department of Pharmacy Practice, Faculty of Pharmacy, Kuwait 15462, Kuwait; zahra.alsairafi@hsc.edu.kw; 5Clinical Laboratory, Amiri Hospital Kuwait, Kuwait 15462, Kuwait; a5_dr_ashraf@hotmail.com; 6Genetics and Bioinformatics, Dasman Diabetes Institute, Kuwait 15462, Kuwait; hamad.ali@dasmaninstitute.org (H.A.); alphonse.thangavel@dasmaninstitute.org (T.A.T.); fahd.almulla@dasmaninstitute.org (F.A.-M.); 7Department of Medical Laboratory Sciences, Faculty of Allied Health Sciences, Health Sciences Center, Kuwait University, Kuwait 15462, Kuwait; 8Medical Division, Dasman Diabetes Institute, Kuwait 15462, Kuwait; ebaa.alozairi@dasmaninstitute.org; 9Diabetology Unit, Amiri Hospital, Dasman Diabetes Institute, Kuwait 15462, Kuwait; abdul_2345@yahoo.com

**Keywords:** diabetic nephropathy, ANGPTL8, IGFBP4, IGF-binding proteins, type 2 diabetes

## Abstract

Diabetic nephropathy (DN) is a complicated condition related to type 2 diabetes mellitus (T2D). ANGPTL8 is a hepatic protein highlighted as a risk factor for DN in patients with T2D; additionally, recent evidence from DN studies supports the involvement of growth hormone/IGF/IGF-binding protein axis constituents. The potential link between ANGPTL8 and IGFBPs in DN has not been explored before. Here, we assessed changes in the circulating ANGPTL8 levels in patients with DN and its association with IGFBP-1, -3, and -4. Our data revealed a significant rise in circulating ANGPTL8 in people with DN, 4443.35 ± 396 ng/mL compared to 2059.73 ± 216 ng/mL in people with T2D (*p* < 0.001). Similarly, levels of IGFBP-3 and -4 were significantly higher in people with DN compared to the T2D group. Interestingly, the rise in ANGPTL8 levels correlated positively with IGFBP-4 levels in T2DM patients with DN (*p* < 0.001) and this significant correlation disappeared in T2DM patients without DN. It also correlated positively with serum creatinine and negatively with the estimated glomerular filtration rate (eGFR, All < 0.05). The area under the curve (AUC) on receiver operating characteristic (ROC) analysis of the combination of ANGPTL8 and IGFBP4 was 0.76 (0.69–0.84), *p* < 0.001, and the specificity was 85.9%. In conclusion, our results showed a significant increase in ANGPTL8 in patients with DN that correlated exclusively with IGFBP-4, implicating a potential role of both proteins in the pathophysiology of DN. Our findings highlight the significance of these biomarkers, suggesting them as promising diagnostic molecules for the detection of diabetic nephropathy.

## 1. Introduction

Type 2 diabetes (T2D) is a major endemic health problem that affects millions of people worldwide [1], whereby insulin resistance is a key underlying mechanism linked to several macro- and microvascular complications, such as cardiovascular disease and diabetic nephropathy (DN) [2,3,4]. Insulin resistance leads to a rise in hepatic glucose secretion and a reduced muscle glucose uptake, which consequently triggers pancreatic β-cells to boost insulin production [5]. The chronic upregulation of insulin production has detrimental effects on pancreatic β-cells’ activity and viability, ultimately leading to diabetes. In addition to T2D, 30–40% of people with T2D develop DN [6,7,8].

The hepatic angiopoietin-like protein 8 (ANGPTL8) has an interesting role in lipid metabolism by inhibiting lipoprotein lipases to regulate triglyceride levels, and it also contributes to the enhancement of glucose tolerance [9,10,11,12,13,14,15]. In a previous study, we reported a significant rise in the levels of circulating ANGPTL8 in people with T2D compared to those without T2D, which was strongly associated with insulin resistance. The level of circulating ANGPTL8 has been associated with atherogenic lipid profiles in high-risk cohorts with T2D or cardiovascular disease [13,14,16]. Elevated serum ANGPTL8 levels were associated with higher all-cause mortality risk in a Chinese population with T2D [17]. In addition to this, Yang et al. suggested a potential role for ANGPTL8 in the pathogenesis of albuminuria in T2D due to the strong link between ANGPTL8 and albuminuria [18].

On the other hand, growth hormone/insulin-like growth factor (IGF)/IGF-binding protein (BP) axis constituents are known to play key roles in both maintaining normal renal function and renal dysfunction that leads to DN [19]. The activity of IGF-1 increases in the diabetic kidney in an autocrine/paracrine style that mediates matrix production, mesangial cell propagation, and movement [20,21]. This process is regulated by IGFBPs both through IGF binding and by IGF-independent mechanisms. As such, the overexpression of IGFBP-1 in T2D was shown to contribute to glomerulosclerosis [22]. High glucose levels were also found to elevate renal IGFBP-3 production, which is a significant mediator of podocyte apoptosis, whereby the loss of podocytes was strongly correlated with albuminuria [23]. The specific mechanisms by which insulin and glucose facilitate increases in IGF-1 and IGFBPs in the kidney, leading to the development of DN, is an area of intensive research [24,25,26].

Currently, the correlation between changes in the levels of ANGPTL8 and DN and its underlying mechanisms are poorly understood. Therefore, the present study investigated the association between DN and circulating ANGPTL8 in people with T2D, compared to people with diabetic nephropathy. Additionally, we investigated the association between ANGPTL8 and IGFBP-1, -3, and -4 and their potential interplay in the pathogenesis of DN.

## 2. Results

This study involved a total of 86 Kuwaiti people that were grouped into 37 individuals with T2D and 49 participants with DN. The demographic, clinical, and biochemical characteristics of the study population are detailed in Table 1. The study groups were age- and body mass index (BMI)-matched (*p* > 0.05).

### 2.1. Elevated Levels of Circulating IGFBP-1, -3, -4, and ANGPTL8 in People with Diabetic Nephropathy

People with T2D and DN showed increased levels of circulating IGFBP-1. However, the difference was not significantly different between people with DN and the T2D group (Figure 1A). On the other hand, IGFBP-3 levels showed a significant increase in people with DN compared to people with T2D (*p* = 0.02, Figure 1B). People with DN showed a significant rise in circulating IGFBP-4 levels (627.8 ± 92.80 ng/mL) compared to people with T2D (278.9 ± 48.71 ng/mL, *p* = 0.003) (Figure 1C). Similarly, circulating ANGPTL8 was significantly higher in people with DN (4443 ± 396.3 ng/mL, *p* < 0.0001) compared to people with T2D (2060 ± 216.4 ng/mL) (Figure 1D).

### 2.2. Elevated IGFBP-4 and ANGPTL8 Are Correlated with Clinical Parameters of Nephropathy

To examine the significance of increased IGFBP-4 and ANGPTL8 levels in DN context, we performed Pearson’s correlation analysis to test the link between the clinical parameters of DN and both IGFBP-4 (Table 2) and ANGPTL8 (Table 3) in our study population. Our data presented a significant positive correlation between the rise in plasma IGFBP-4 and serum creatinine levels (r = 0.703, *p* < 0.001, Figure 2A) in people with DN. Additionally, circulating IGFBP-4 was negatively correlated with eGFR (r = −0.55, *p* < 0.001, Figure 2B). IGFBP-4 and urine creatinine showed no significant correlation (Figure 2C). Similarly, there was a positive correlation between circulating ANGPTL8 and serum creatinine in people with DN (r = 0.37, *p* = 0.007, Figure 3A), while ANGPTL8 was negatively correlated with both eGFR (r = −0.44, *p* = 0.002, Figure 3B) and urine creatinine (r = −0.407, *p* = 0.004, Figure 3C).

### 2.3. ANGPTL8 Is Correlated with IGFBP-4 in Nephropathy

To study the link between ANGPTL8 and IGFBPs, we utilised Spearman’s correlation analysis, which showed a significant positive correlation between ANGPTL8 and IGFBP-4 in people with DN (r = 0.468, *p* < 0.001, Table 3). ANGPTL8 was correlated with IGFBP-1 in people with T2D (r = 0.325, *p* = 0.049, Table 3), but there was no association with IGFBP-3 in our study population.

### 2.4. Predictive Parameters of ANGPTL8 Increase in Nephropathy

To further test the link between ANGPTL8 and other biochemical parameters, we performed multiple stepwise regression analysis with a number of predictors (Table 4). Our model presented IGFBP4 as a predictive marker with a significant positive regression weight (β = 0.435, *p* = 0.002, Table 4) for the elevation of ANGPTL8 in people with DN. This marker showed an independent correlation with ANGPTL8 levels (F_2,43_ = 10.783, *p* < 0.001, r^2^ = 33%). On the other hand, people with T2D presented both IGFBP4 (β = 0.624, *p* = 0.006) and VLDL (β = 0.413, *p* = 0.009) as predictors for an increase in circulating ANGPTL8 (Table 4), while LDL acted as a negative predictor of ANGPTL8 increase in people with T2D (β = −0.355, *p* = 0.023, Table 4). Collectively, these markers independently correlated with the increase in ANGPTL8 levels (F_3,34_ = 6.233, *p* = 0.005, r^2^ = 27%). To sum up, our data presented IGFBP4 as a marker and significant positive predictor with ANGPTL8 as a dependent variable in people with T2D with and without nephropathy. To identify the cut-off value for ANGPTL8 and IGFBP4 and to further evaluate the predictive accuracy of these biomarkers for people with DN, we employed the ROC curves analysis (Figure 4). The analysis showed that the area under the curve (AUC) (95% CI) was 0.73 (0.65–0.81) for ANGPTL8, 0.74 (0.66–0.82) for IGFBP4, and 0.76 (0.69–0.84) for their combination. The optimal cut-off value for predicting DN in all populations with ANGPTL8 exceeded 839.82 ng/mL with 84.1% sensitivity and a specificity of 80.4%. The optimal cut-off value for using IGFBP4 as a predictive marker for DN was higher than 230.05 ng/mL with 67.6% sensitivity and a specificity of 58.5%. The specificity of the combination of ANGPLT8 and IGFBP4 was increased at 85.9%.

## 3. Discussion

In the current study, we measured levels of circulating ANGPTL8 and IGFBP-1, -3, and -4 in a Kuwaiti Arab cohort of patients with T2D and DN. Our results disclosed a positive association between elevated levels of ANGPTL8 and IGFBP-4 in people with DN. The significant increase in the levels of both proteins in patients with DN compared to people with T2D suggested a prospective role for both ANGPTL8 and IGFBP-4 in DN. Additionally, the ROC curve analysis suggested that both proteins are closely associated with DN. This emphasised their importance as potential diagnostic biomarkers for DN in people with T2D.

DN is a main microvascular diabetic complication that is characterised by the thickening of glomerular and tubular membranes and an augmented mesangial matrix, which eventually advance to glomerulosclerosis and tubulointerstitial fibrosis [27,28,29,30,31]. People with T2D and microalbuminuria show a rise in renal IGF-1 levels, IGFBP-3 protease activity, and the increased excretion of bioactive growth hormone, IGF-1, and IGFBP-3 proteins that indicate a defective growth hormone/IGF/IGFBP system [32]. Our data presented a significant increase in circulatory IGFBP-4 in people with DN compared to people with T2D. IGFBP-4 is predominantly produced in the glomerulus. Therefore, higher levels of IGFBP-4 might be due to increased secretion, reduced degradation, or weakened glomerular filtration. This protein is broadly expressed in vivo, most abundantly in the kidney, and is involved in both physiological and pathological mechanisms [33]. Earlier reports have shown that a rise in circulating IGFBP-4 correlates with the severity of chronic renal alteration in adults and children [34]. Additionally, a recent study reported a significant upregulation of IGFBP5 levels in diabetic kidney disease (DKD), although the exact function of IGFBP5 and its role in DKD remains elusive [35]. In keeping with previous findings, the present study showed a significant association between IGFBP-4 levels and eGFR, implicating a potential role in DN.

On the other hand, ANGPTL8 and ANGPTL3 regulate plasma triglyceride levels by inhibiting lipoprotein lipases [36]. The upregulation of circulating ANGPTL8 in people with T2D was reported by several studies [13,17,37,38,39,40,41]. In agreement with the reported ANGPTL8 increase in patients with T2D [42], we previously reported a three-fold increase in circulating ANGPTL8 in people with T2D compared to the control [13]. In the current study, people with DN showed a significant rise in ANGPTL8 compared to those with T2D. Furthermore, elevated levels of ANGPTL8 positively correlated with serum creatinine and showed a negative correlation with eGFR and urine creatinine in people with DN. This was in keeping with a previous report where different stages of albuminuria showed different levels of ANGPTL8, whereby ANGPTL8 was suggested as a novel regulator in developing DN [18,43] and a novel predictive risk marker for all-cause mortality in people with T2D [17]. In another report involving a group of newly diagnosed patients with T2D, ANGPTL8 was highlighted as a potential predictive marker for diabetic complications, in particular, DN and deteriorated kidney functions [44].

In addition to the significant increase in ANGPTL8 levels, people with DN had microalbuminuria (592.87 ± 226.96 mg/day, p = 0.01), which is in agreement with the reported link between circulating ANGPTL8 and albuminuria [43]. The rise in ANGPTL8 production was attributed to losing albumin, which causes insulin resistance and increased need for insulin in people with T2D and albuminuria. In previous work by Ebert et al., the rise in ANGPTL8 correlated with clinical and biochemical measures of renal function in a group of patients with diabetes and renal dysfunction [45]. The proposed mechanism for the increase in ANGPTL8 production in DN was dysregulated lipid metabolism, where ANGPTL8 was introduced as a novel endocrine regulator contributing to the development of DN [43]. On the other hand, levels of ANGPTL8 were significantly reduced at a state of renal failure in patients with chronic haemodialysis; nonetheless, the regulatory mechanism causing the elimination of ANGPTL8 in end-stage renal disease is still obscure [45].

Previous studies have shown a correlation between low levels of IGFBP-1 and hyperinsulinemia [46]. However, the IGFBP-1 level is known to increase with the progression of T2D despite persistent hyperinsulinemia due to a progressive hepatic insulin resistance [47,48,49]. Similarly, IGFBP-3 induced mesangial apoptosis in the presence of high ambient glucose or tumour necrosis factor-α in the kidney [32]. DN was accompanied by IGFBP-3 proteolysis in urine, whereby the resultant IGFBP-3 fragments contributed to raising albumin secretion through increasing the apoptosis of glomerular epithelial cells [32]. In this report, the elevation in ANGPTL8 correlated positively with IGFBP-4, and this occurred exclusively in people with DN, thus supporting a possible ANGPTL8–IGFBP4 interplay that contributes to the pathogenesis and progression of DN. This finding implicates the potential for using the increase in circulating ANGPTL8 and IGFBP4 as a diagnostic tool for the early detection of nephropathy progression in patients with T2D. Due to the well-known function of ANGPTL8 in lipid regulation, it is assumed that its persistent increase results in dyslipidaemia and increased insulin resistance, which aggravates metabolic stress and ultimately leads to the development of metabolic diseases like T2D and complications such as DN. One of the main limitations of the current study is its cross-sectional design, which made it difficult to establish the biological role of these proteins and their potential contribution to the pathophysiology of DN. Thus, future research should involve prognostic studies to validate the relationship between ANGPTL8 and IGFBP4 in DN. This would also establish an understanding of the cause–effect relationship between DN and these proteins.

In conclusion, our data mainly highlight the presence of a significant positive correlation between ANGPTL8 and IGFBP-4 in people with DN. There was a significant increase in circulating ANGPTL8 and IGFBP-4 in people with DN, and both proteins exhibited a significant correlation with clinical indicators of DN. The rise in ANGPTL8 and IGFBP-4 correlated positively with serum creatinine and negatively with eGFR, which suggested their potential role in DN. With the established link between ANGPTL8 and renal dysregulation, the presence of a positive correlation between ANGPTL8 and IGFBP-4 emphasised a possible interplay between these proteins that contributes to DN progression and/or severity. ROC analysis of the combination of ANGPTL8 and IGFBP4 stresses the importance of these proteins as promising diagnostic biomarkers for DN in patients with T2D. Future studies are needed to elucidate the link between ANGPTL8 and IGFBP4 and the mechanism of action through which they contribute to diabetes and its complications.

## 4. Materials and Methods

### 4.1. Study Population

A total of 86 participants were enrolled in the study, and this involved 37 participants with type 2 diabetes (T2D) and 49 subjects with diabetic nephropathy (DN) that were age- and body mass index (BMI)-matched. The participants were recruited at Dasman Diabetes Institute (Dasman, Kuwait), and were enrolled in the study after giving written informed consent. The study protocol was approved by the Ethical Review Board of Dasman Diabetes in accordance with the ethical guidelines outlined in the Declaration of Helsinki. People with T1D, renal transplant, or end-stage renal disease were excluded from the study.

### 4.2. Study Groups Definitions

People were diagnosed with T2D by showing persistent hyperglycaemia with fasting blood glucose (FBG) level > 7 mmol/L and OGTT 2-h blood glucose > 11 mmol/L and presenting with normal kidney function. A clinical diagnosis of DN was determined by a nephrologist following the American Diabetes Association criteria [50], and this involved showing pronounced T2D and persistent elevation in ACR > 30 mg/g.

### 4.3. Anthropometric and Biochemical Measurements

The whole-body composition of all participants was determined by dual-energy radiographic absorptiometry (Lunar DPX, Lunar radiation, Madison, WI, USA). Blood pressure readings were an average of three measurements taken by an Omron HEM-907XL digital sphygmomanometer, with a 5 to 10 min rest between each reading. Fasting blood samples were collected into vacutainer-EDTA tubes and centrifuged at 400× *g* for 10 min to separate the plasma, which was aliquoted and stored at −80 °C until assayed. Fasting blood glucose (FBG), triglyceride (TG), total cholesterol (T. Chol), low-density lipoprotein (LDL), and high-density lipoprotein (HDL) levels were measured on a Siemens Dimension RXL chemical analyser (Diamond Diagnostics, Holliston, MA, USA). Levels of haemoglobin A1c (HbA1C) were determined using a VariantTM device (Bio-Rad, Hercules, CA, USA). Urine samples collected from the first morning void were obtained from all participants to determine the concentrations of albumin, creatinine, and albumin-to-creatinine ratio (ACR) using a CLINITEK Novus Automated Urine Chemical Analyzer (Siemens Healthineers, Erlangen, Germany).

### 4.4. Levels of Serum and Urinary Creatinine and Urinary Protein

Urinary protein concentrations were determined using a Coomassie Plus protein assay kit (Pierce, Rockford, IL, USA) according to the manufacturer’s instructions. Urinary and serum creatinine concentrations were determined with a VITROS 250 automatic analyser (New York, NY, USA). The estimated glomerular filtration rate (eGFR) was calculated using the Modification of Diet in Renal Disease study equation.

### 4.5. ANGPTL8 Enzyme-Linked Immunosorbent Assay (ELISA)

Levels of ANGPTL8 protein were quantified in plasma as described previously [13,51]. Frozen plasma was thawed on ice followed by five minutes of centrifugation at 10,000× *g* at 4 °C to remove any debris, avoiding repeated freeze–thaw cycles. ANGPTL8 plasma levels were determined using an ELISA kit (Cat. No. E1164H, Wuhan EIAAB Science Co., Wuhan, China), as described previously [13,51]. We found no significant cross reactivity with other proteins, and the intra- and inter-assay coefficients of variation were 2.1–4.6% and 7.3–9.6%, respectively.

### 4.6. Quantifying IGFBP-1, -3, and -4

Levels of IGFBP-1, -3, and -4 were measured by a Magnetic Luminex Assay kit (R&D Systems Europe, Ltd., Abingdon, UK) according to the manufacturer’s protocol. The assay is composed of premixed microparticles and biotinylated detection antibodies that enable the screening of multiple human biomarkers in a single test.

### 4.7. Statistical Analysis

Both study groups—people with T2D and DN—were compared using an unpaired Student’s *t*-test with *p* < 0.05 to determine statistical significance. All data were reported as mean ± standard error of the mean (SEM). Pearson’s correlation coefficients were estimated to assess the association of ANGPTL8 with ANGPTL3, ANGPTL4, and IGFBP-1, -3, and -4. Multiple regression analysis was implemented to identify parameters independently associated with ANGPTL8. Statistical assessments were two-sided and considered to be significant at *p* < 0.05. The area under the receiver operating characteristic (ROC) curve was calculated to identify the cut-off value for ANGPTL8 and IGFBP4 as biomarkers for DN in all populations. All analyses were performed using SAS version 9.2 software (SAS Institute, Cary, NC, USA).

## Figures and Tables

**Figure 1 ijms-24-14244-f001:**
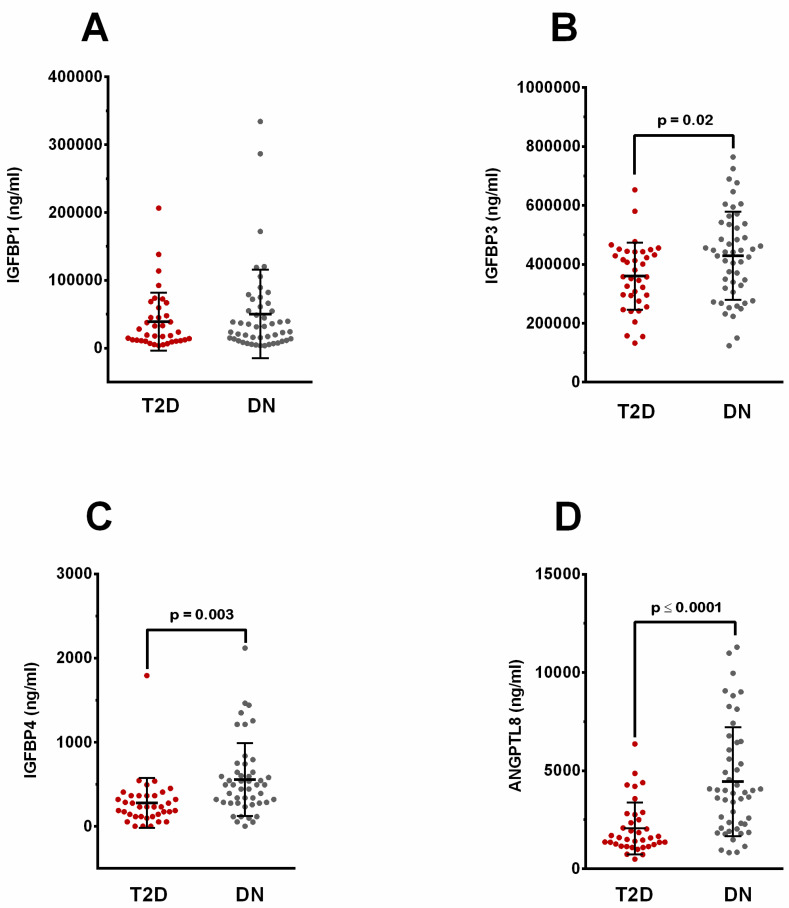
Circulating plasma levels of IGFBP-1, -3, -4, and ANGPTL8 in people with T2D (n = 37) and DN (n = 49). (**A**) People with DN showed higher levels of IGFBP-1 compared to people with T2D. (**B**) IGFBP-3 levels were higher in people with DN compared to people with T2D, and the difference was significantly different (*p* = 0.02). (**C**) Circulating levels of IGFBP-4 were significantly higher in people with DN in comparison to people with T2D (*p* = 0.003). (**D**) Levels of ANGPTL8 were elevated in people with DN and the difference was significant compared to people with T2D (*p* < 0.001).

**Figure 2 ijms-24-14244-f002:**
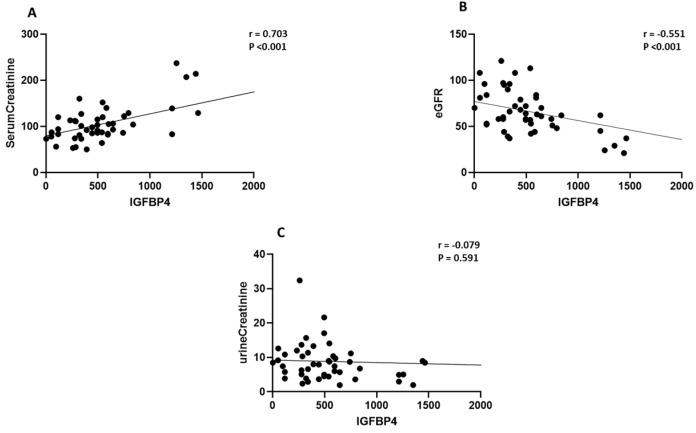
Correlation analysis between IGFBP-4 and clinical variables associated with DN. Pearson’s correlation coefficient showed a significant (**A**) positive correlation between serum creatinine and IGFBP-4 (r = 0.703, *p* < 0.001) and (**B**) negative correlation between eGFR and IGFBP-4 (r = −0.551, *p* < 0.001), but (**C**) the correlation between urine creatinine and IGFBP4 was not significant (r = −0.079, *p* = 0.59).

**Figure 3 ijms-24-14244-f003:**
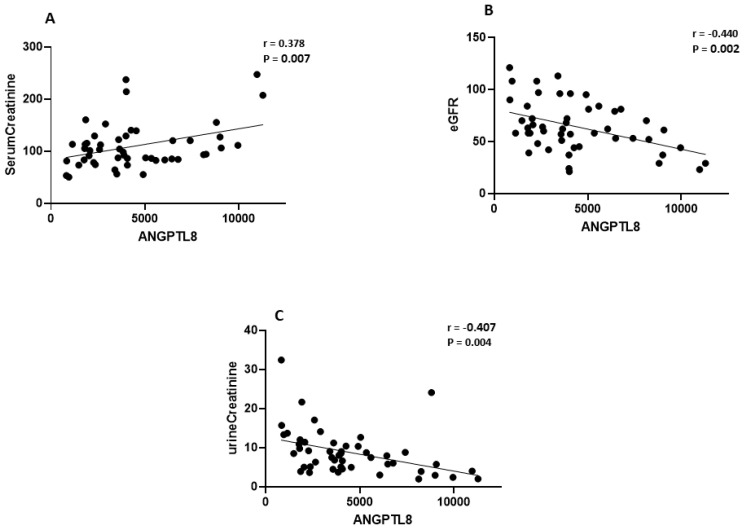
Correlation analysis between ANGPTL8 and clinical variables associated with DN. Pearson’s correlation coefficient showed a significant (**A**) positive correlation between serum creatinine and ANGPTL8 (r = 0.378, *p* = 0.007) and negative correlation between (**B**) eGFR and ANGPTL8 (r = −0.44, *p* = 0.002) and (**C**) urine creatinine with ANGPTL8 (r = −0.407, *p* = 0.004).

**Figure 4 ijms-24-14244-f004:**
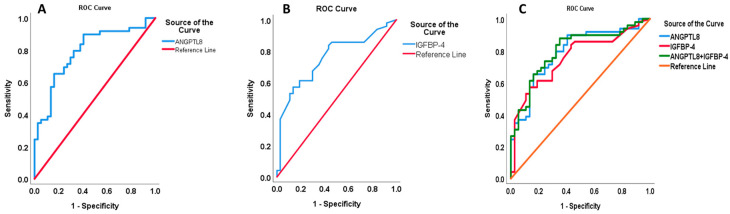
ROC curve analysis performed to identify the cut-off value of ANGPTL8 and IGFBP4 as biomarkers for DN vs. T2DM. (**A**) AUC for ANGPTL8 (0.79 (0.70–0.89) *p* < 0.001). (**B**) AUC for IGFBP4 (0.76 (0.65–0.86) *p* < 0.001). (**C**) AUC of the combination, ANGPTL8 and IGFBP4 (0.80 (0.71–0.90) *p* < 0.001).

**Table 1 ijms-24-14244-t001:** Anthropometric and clinical characteristics of participants with T2D and DN.

Parameters	T2D n = 37	DN n = 49	*p*
Gender (M/F)	16/21	34/15	0.015
Age (years)	57.27 ± 1.11	59.0 ± 1.32	0.319
BMI (kg/m^2^)	31.56 ± 0.60	31.88 ± 0.64	0.712
SBP (mmHg)	131.6 ± 2.46	130.98 ± 5.36	0.975
DBP (mmHg)	72.46 ± 1.78	68.51 ± 3.01	0.262
Fasting Glucose (mmol/L)	7.86 ± 0.41	8.67 ± 0.44	0.177
HbA1C (%)	7.59 ± 0.19	7.87 ± 0.25	0.371
TC (mmol/L)	4.10 ± 0.16	3.87 ± 0.12	0.258
TGL (mmol/L)	1.39 ± 0.20	1.70 ± 0.13	0.211
HDL (mmol/L)	1.21 ± 0.6	1.10 ± 0.04	0.084
LDL (mmol/L)	2.28 ± 0.13	2.01 ± 0.11	0.099
VLDL (mmol/L)	0.56 ± 0.08	0.68 ± 0.05	0.216
C Peptide (pg/mL)	0.67 ± 0.05	0.66 ± 0.06	0.955
Serum Creatinine (mg/L)	73.08 ± 3.14	109.57 ± 6.28	<0.001
eGFR (mL/min/1.73 m^2^)	86.64 ± 3.03	64.08 ± 3.48	<0.001
Albumin (mcg/L)	39.11 ± 0.50	37.51 ± 0.53	0.031
CRP (μg/mL)	0.38 ± 0.07	0.46 ± 0.09	0.505
ACR (mg/L)	10.24 ± 1.12	707.07 ± 217.78	0.002
Urine Creatinine (mg/day)	12.62 ± 1.07	8.78 ± 0.84	0.006
Microalbumin (mg/day)	13.57 ± 1.56	592.87 ± 226.96	0.014

Data are mean ± standard error of the mean; SBP: systolic blood pressure; DBP: diastolic blood pressure; ACR: albumin/creatinine ratio.

**Table 2 ijms-24-14244-t002:** Pearson’s correlation analysis for IGFBP-4 in all study groups: T2D and DN.

	IGFBP-4			
Parameters	T2D		DN	
	r	*p*	r	*p*
Age (years)	0.117	0.49	0.3	0.037
BMI (kg/m^2^)	0.156	0.355	−0.072	0.629
SBP (mmHg)	0.019	0.913	0.067	0.646
DBP (mmHg)	−0.278	0.096	0.058	0.691
Fasting Glucose (mmol/L)	−0.118	0.488	0.394	0.005
HbA1C (%)	−0.035	0.836	−0.159	0.275
T. Chol (mmol/L)	−0.001	0.996	−0.334	0.019
TGL (mmol/L)	0.001	0.994	0.197	0.175
HDL (mmol/L)	−0.039	0.818	−0.096	0.511
LDL (mmol/L)	0.009	0.958	−0.482	0.001
VLDL (mmol/L)	0.001	0.994	0.197	0.175
C Peptide (pg/mL)	−0.098	0.562	−0.039	0.789
Serum Creatinine (mg/L)	0.017	0.92	0.703	<0.001
eGFR (mL/min/1.73 m^2^)	−0.148	0.388	−0.551	<0.001
Albumin(mcg/L)	−0.177	0.294	−0.259	0.073
ACR (mg/g)	0.07	0.679	0.211	0.146
Urine Creatinine (mg/day)	0.217	0.196	−0.079	0.591
Microalbumin (mg/day)	0.288	0.084	0.05	0.731
IGFBP1	0.058	0.732	0.08	0.586
IGFBP3	−0.214	0.204	0.187	0.198
ANGPTL8	0.258	0.124	0.468	0.001

r, Pearson’s correlation coefficient with significance at *p* < 0.05.

**Table 3 ijms-24-14244-t003:** Pearson’s correlation analysis for ANGPTL8 in people with T2D and DN.

	ANGPTL8			
Parameters	T2D		DN	
	r	*p*	r	*p*
Age (years)	0.333	0.044	0.378	0.007
BMI (kg/m^2^)	0.068	0.687	−0.137	0.358
SBP (mmHg)	−0.041	0.808	−0.134	0.357
DBP (mmHg)	−0.331	0.046	−0.246	0.088
Fasting Glucose (mmol/L)	−0.091	0.593	0.016	0.915
HbA1C (%)	−0.145	0.391	−0.223	0.123
T. Chol (mmol/L)	−0.22	0.191	−0.228	0.115
TGL (mmol/L)	0.075	0.66	0.022	0.879
HDL (mmol/L)	−0.086	0.614	−0.114	0.436
LDL (mmol/L)	−0.323	0.054	−0.242	0.098
VLDL (mmol/L)	0.077	0.652	0.023	0.878
C Peptide (pg/mL)	0.232	0.167	0.085	0.559
Serum Creatinine (mg/L)	0.228	0.176	0.378	0.007
eGFR (mL/min/1.73 m^2^)	−0.372	0.026	−0.44	0.002
Albumin(mcg/L)	−0.03	0.86	−0.182	0.21
ACR (mg/g)	0.007	0.969	0.075	0.609
Urine Creatinine (mg/day)	0.07	0.682	−0.407	0.004
Microalbumin (mg/day)	0.116	0.495	−0.06	0.684
IGFBP1	0.325	0.049	0.069	0.636
IGFBP3	0.203	0.227	0.211	0.145
IGFBP4	0.258	0.124	0.468	0.001

r, Pearson’s correlation coefficient with significance at *p* < 0.05.

**Table 4 ijms-24-14244-t004:** Multiple regression analysis to identify parameters associated with ANGPTL8.

Parameters	T2D		DN	
	β	*p*-Value	β	*p*-Value
VLDL	**0.413**	**0.009**	−0.037	0.982
LDL	**−0.355**	**0.023**	−0.004	0.775
HDL	−0.022	0.740	−0.07	0.991
IGFBP4	**0.624**	**0.006**	**0.435**	**0.002**

## Data Availability

All data relevant to the study are included in the article. Further data are available upon reasonable request.

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
