# Peer review of "Increased Levels of Circulating IGFBP4 and ANGPTL8 with a Prospective Role in Diabetic Nephropathy"

_ijms, 2023, doi:10.3390/ijms241814244_

Round 1

Reviewer 1 Report

1. What was the criteria of DN diagnosis? In my opinion the presence of albuminuria in patient with T2D is not enough. There might be several reasons of proteinuria among diabetic patients, not necessarily related to DN.

2. Including healthy control group was not appropriate. While testing markers involved in T2D pathway, what was the idea of including patients without T2D. There is a simple and robust marker distinguishing T2D patient from the healthy one - blood glucose. Authors should exclude healthy patients and recalculate the results comparing only those with T2D and DN. 

3. If serum creatinine is rising then eGFR is plummeting, there is no need to focus on both. 

The English language used is fine and needs only small corrections.

Author Response

We would like to thank the reviewer for positive comments and feedback, and we appreciate the chance to respond to the raised points, hoping that our responses and explanation would be satisfactory.

  1. What was the criteria of DN diagnosis? In my opinion the presence of albuminuria in patient with T2D is not enough. There might be several reasons of proteinuria among diabetic patients, not necessarily related to DN.

We thank the reviewer for this question. In the current study as it is mentioned in the materials and methods section, people were diagnosed with DN by a nephrologist following the American Diabetes Association criteria, which involved presenting a pronounced T2D (i.e. FBG > 7 mmol/L and 2-hr OGTT > 11mmol/L) and persistent elevation in ACR > 30 mg/g. The rise in albuminuria levels and proteinuria was not the biomarker used to diagnose or classify people with DN. We’ve presented these biomarkers in the data tables as indicators/signs reflecting DN in the study group.    

  1. Including healthy control group was not appropriate. While testing markers involved in T2D pathway, what was the idea of including patients without T2D. There is a simple and robust marker distinguishing T2D patient from the healthy one - blood glucose. Authors should exclude healthy patients and recalculate the results comparing only those with T2D and DN. 

We appreciate the reviewer’s comment and concerns accordingly, we have addressed this issue and healthy group data was removed from the analysis. The paper/data is modified to present data from people with T2D and DN, comparing the levels of IGFBPs, ANGPTL8 and other analytes. This modification is presented in Tables 1-4, figure 1 and throughout the paper.

  1. If serum creatinine is rising then eGFR is plummeting, there is no need to focus on both.

We thank the reviewer for comment, and we fully acknowledge that people with DN are characterized by persistent albuminuria (or albuminuria excretion rate of >300 mg/d) with a progressive decline in glomerular filtration rate (GFR). Therefore, these were simply listed with the data collected from the study participants and they were not intended to highlight a new finding.   

Reviewer 2 Report

The paper is clear and interesting, however some condieration on the effects of current antidiabetics drug on ANGPTL8 levels, and speculation on the relevance on the clinical effect in preventing DN would add important considerations.

Author Response

We would like to thank the reviewer for the positive comments and feedback, and we appreciate the chance to respond to the raised points, hoping that our responses and explanation will be satisfactory.

We highly acknowledge the concern of the reviewer, and considering the potential effects of antidiabetic medications on ANGPTL8 levels is a valid concern and a matter that we are planning to look closely into in future studies. However, considering the nature of the current study, which has a cross-sectional nature, we can’t conclude that the difference in ANGPTL8 levels between people with T2D and DN was due to the effect of the antidiabetic medications. Since this study has more of a descriptive nature and we can not, with the current setup, determine cause-and-effect relationship between the various variables. Further research and experimentation to scrutinize the effect of antidiabetic medication and the combined involvement of ANGPTL8 and IGFBP4 in DN are in our plans for future studies.

Reviewer 3 Report

In the present study authors investigated whether blood levels of ANGPTL8 and IGFBP-4 are possible predictors of diabetic nephropathy (DN) among T2DM patients. The paper is well written, but several concerns rose during the review process. 

First, IGFBP-4 is not a novelty as the author's already published it recently (BMC Nephrol, 2022), where IGFBP-4 correlated positively also with creatinine level and negatively with eGFR. However, this BMC Nephrol paper is very similar to the present proposal. Even the structure of tables, figures, correlation plots as well as ROC curves look almost identical (!). The investigated patient groups are also similar (control, T2D and DN). Still, in this previous work, IGFBP-4 failed to show any correlation with albuminuria in DN, in contrast to the present analysis. This implies that IFGBP-4 shows a study-dependent correlation only, and is not a robust biomarker for DN. 

Second, the fasting glucose of DN patients is almost 1 mmol/l higher than of T2D, this could also bias the results. Circulating ANGPTL8 levels were independently associated with fasting blood gluose levels in T2DM patients (J Clin Endocrinol Metab. 2014 Dec;99(12):E2510-7). However, in contrast to the proposed study, they found that renal function did not influence ANGPTL8 levels, but insulin induced its secretion. Thus, as DN patients here clearly had higher blood glucose levels, this presumably promoted higher insulin secretion as well. 

Minor issues:

1) In Table 1, SBP of T2D group is 13.16, this should be a typo.

2) Figure 1 c,d shows same ANPTL8 and IGFBP4 results as the end of table 1, and again within the text (Line 104-108) . This is unnecessary repeat, delete from table and from the text.

3) Regarding statistical analysis, please recalculate ANOVA by using more powerful post-hoc test, such as Tukey. Bonferroni has the lowest statistical power.

4) Chart format on Figure 1 should be changed from bar charts to scatter-plot with mean +/- SD, in order to see individual data.

5) Line 271: what does 2-hour fasting blood glucose level mean? I suppose you meant OGTT 2-hr value, but this is far not fasting condition! 

Author Response

We would like to thank the reviewer for his comments and feedback, and we appreciate the chance to respond to the raised points, hoping that our responses and explanation would be satisfactory.

  1. “First, IGFBP-4 is not a novelty as the author's already published it recently (BMC Nephrol, 2022), where IGFBP-4 correlated positively also with creatinine level and negatively with eGFR. However, this BMC Nephrol paper is very similar to the present proposal. Even the structure of tables, figures, correlation plots as well as ROC curves look almost identical (!). The investigated patient groups are also similar (control, T2D and DN). Still, in this previous work, IGFBP-4 failed to show any correlation with albuminuria in DN, in contrast to the present analysis. This implies that IFGBP-4 shows a study-dependent correlation only, and is not a robust biomarker for DN”.

We highly appreciate and acknowledge the reviewer’s comment regarding the novelty of IGFBP4, specifically with reference to our previous publication. As it was highlighted by the reviewer, we have reported IGFBP4 as a novel potential biomarker for DN in a previous publication, and our intention is not to duplicate reporting this idea through the current report. At the current time, we are focused on the combined correlation linking both IGFBP4 and ANGPTL8 to DN. Additionally, we have taken the comment of the reviewer with great consideration and modified the analysis to reflect a comparison between people with T2D and DN, where healthy participants were excluded from the analysis.

  1. Second, the fasting glucose of DN patients is almost 1 mmol/l higher than of T2D, this could also bias the results. Circulating ANGPTL8 levels were independently associated with fasting blood gluose levels in T2DM patients (J Clin Endocrinol Metab. 2014 Dec;99(12):E2510-7). However, in contrast to the proposed study, they found that renal function did not influence ANGPTL8 levels, but insulin induced its secretion. Thus, as DN patients here clearly had higher blood glucose levels, this presumably promoted higher insulin secretion as well. 

We Appreciate the reviewer point. However, the referred study assessed the level of ANGPTL8 in 62 patients with T2DM as compared with 58 nondiabetic subjects. Within both groups, about half of the patients were on chronic hemodialysis. The cohort setting and size are different for any comparison. Further, Yang et al. found that serum ANGPTL8 levels were significantly increased in type 2 diabetic patients with albuminuria and suggested of potential role of ANGPTL8 in the pathogenesis of albuminuria in type 2 diabetes. In this manuscript, we are focused on the combined correlation linking both IGFBP4 and ANGPTL8 to DN. Our results showed a significant increase in ANGPTL8 in patients with DN that correlated exclusively with IGFBP-4, implicating a potential role of both proteins in the pathophysiology of DN.

References:

Yang L, Song J, Zhang X, Xiao L, Hu X, Pan H, Qin L, Liu H, Ge B, Zheng T. Association of Serum Angiopoietin-Like Protein 8 With Albuminuria in Type 2 Diabetic Patients: Results From the GDMD Study in China. Front Endocrinol (Lausanne). 2018 Jul 18;9:414. doi: 10.3389/fendo.2018.00414. PMID: 30072957

Minor issues:

  1. In Table 1, SBP of T2D group is 13.16, this should be a typo.

We appreciate the reviewer’s diligence, and the typo is now corrected.  

  1. Figure 1 c,d shows same ANPTL8 and IGFBP4 results as the end of table 1, and again within the text (Line 104-108) . This is unnecessary repeat, delete from table and from the text.

We acknowledge the reviewer comment and following this opinion, the values of IGFBP4 and ANGPTL8 were removed from Table 1. Now the values of circulating IGFBP4 and ANGPTL8 are only mentioned in the text and reflected in the plots figure 1 C and D.

  1. Regarding statistical analysis, please recalculate ANOVA by using more powerful post-hoc test, such as Tukey. Bonferroni has the lowest statistical power.

We thank the reviewer for this comment, and this is no longer applicable to the data set since we have excluded the healthy group from the study populations.

  1. Chart format on Figure 1 should be changed from bar charts to scatter-plot with mean +/- SD, in order to see individual data.

We genuinely appreciate the reviewer’s constructive suggestion. We have changed the bar charts to scatter-plot with mean +/- SD for better visualization of the data.

  1. Line 271: what does 2-hour fasting blood glucose level mean? I suppose you meant OGTT 2-hr value, but this is far not fasting condition! 

We appreciate the reviewer’s diligence. Now it is corrected, and it read as follow:

“People were diagnosed with T2D by showing persistent hyperglycemia with fasting blood glucose (FBG) level > 7 mmol/L and 2-hr OGTT > 11 mmol/L and presented normal kidney function.

Round 2

Reviewer 1 Report

I have no further remarks, after implementing my previous comments.

Reviewer 3 Report

The manuscript has been slightly improved. Still, having 2 study groups in statistical analyses, ANOVA with Bonferroni test can not be performed as it is designed for at least 3 groups or more. Moreover, Bonferroni test too often gives significant results having the weakest statistical power among the possible post-hoc tests.

My major concern regarding statistics was therefore not answered. Indicating the precise statistical method is essential in any scientific report. 

**** After checking the corrected version provided by the authors:

"I verified the corrected paper, and although the statistical issues seem to be corrected now, I still believe that the paper's quality does not reach the Journal's merits, and would fit more into a clinical journal with lower impact factor as it does not really contain novelry."

Author Response

Response:

Dear IJMS academic Editor,

We acknowledge the concern of both reviewer and editor; however we assure you that the highlighted point is nothing but a typo which we have unfortunately missed correcting the wording for in section “4.7 Statistical analysis”.

Our data throughout the paper, in particular Table 1, Figure 1 and section 2.1 reflect the modified data following the reviewer’s suggestion to remove the control group and perform a comparison/statistical analysis involving people with T2D and DN, which naturally will be performed using a student’s t-test. For this we assure the academic Editor and the reviewer that the modified data in the manuscript reflects a comparison using an unpaired t-test. This is further confirmed by the new p-values in Table 1, which are different from the previous version (i.e. the initial submitted version) where we had 3-study groups and used ANOVA as a statistical test.

Therefore, we appreciate a chance to reconsider your decision since it was based on having a typo error which we highly apologize for. We hope that our responses and explanation will be satisfactory, and this mistake has been corrected in section 4.7 in the enclosed manuscript.

Sincerely yours,

Jehad Abubaker, PhD.

Principal Scientist & Head of Department

Biochemistry and Molecular Biology at Dasman Diabetes Institute

[email protected]
